# High Serum IL-31 Concentration Is Associated with Itch among Renal Transplant Recipients

**DOI:** 10.3390/jcm11154309

**Published:** 2022-07-25

**Authors:** Piotr K. Krajewski, Kinga Tyczyńska, Klaudia Bardowska, Piotr Olczyk, Danuta Nowicka-Suszko, Dariusz Janczak, Hanna Augustyniak-Bartosik, Magdalena Krajewska, Jacek C. Szepietowski

**Affiliations:** 1Department of Dermatology, Venereology and Allergology, Wroclaw Medical University, 50-368 Wroclaw, Poland; piotr.krajewski@umw.edu.pl (P.K.K.); tyczynska.king@gmail.com (K.T.); klbardowska@gmail.com (K.B.); danuta.nowicka-suszko@umw.edu.pl (D.N.-S.); 2Department of Nephrology and Transplantation Medicine, Wroclaw Medical University, 50-556 Wroclaw, Poland; piotr.olczyk@student.umw.edu.pl (P.O.); hanna.augustyniak-bartosik@umw.edu.pl (H.A.-B.); magdalena.krajewska@umw.edu.pl (M.K.); 3Department of Vascular, General and Transplantation Surgery, Wroclaw Medical University, 50-566 Wroclaw, Poland; dariusz.janczak@umw.edu.pl

**Keywords:** renal transplant recipients, renal transplantation, chronic itch, itch, IL-31

## Abstract

Chronic itch (CI) is a common symptom caused by both dermatological and systemic disorders. CI is also a frequent, burdensome symptom among renal transplant recipients (RTR); however, its pathophysiology is not fully understood. The aim of this study was to assess the differences in concentration of IL-31 among itchy RTR. The study was performed on a group of selected 129 RTRs (54 itchy and 75 non-itchy patients). Itch severity was assessed with the use of the numeral rating scale (NRS) and the 4-item itch questionnaire (4IIQ). Every subject had his blood drawn to measure the concentration of IL-31. The results were subsequently compared and correlated. The mean concentration differed significantly between RTR suffering from itch (602.44 ± 534.5 pg/mL), non-itchy RTR (161.49 ± 106.61 pg/mL), and HC (110.33 ± 51.81 pg/mL) (*p* < 0.001). Post-hoc analysis revealed a statistically significantly increased IL-31 serum concentration in itchy RTR in comparison to the non-itchy RTR group (*p* < 0.001) and HC (*p* < 0.001). No significant difference was observed in IL-31 serum levels between non-itchy RTRs and HC. No correlation between IL-31 and itch intensity was found. The results of our study clearly demonstrate the association between IL-31 levels and CI in patients after renal transplantation.

## 1. Introduction

Chronic itch (CI) is defined as an unpleasant sensation that leads to scratching, of a duration of at least 6 weeks [1]. Although it is the most common symptom in dermatology, it has been also proven that CI affects up to one-fourth of the general population (lifetime prevalence) [2]. CI may also be a sign of systemic disorders including cholestasis, endocrinological and hematological diseases, malignancy, and chronic kidney disease (CKD) [3]. Chronic kidney disease-associated itch (CKD-aI) or uremic itch (UI) is a common symptom in patients suffering from end-stage renal disease treated with hemodialysis (HD), with an overall incidence of moderate to extreme pruritus in 35–44% [4,5,6,7]. CI is also a frequent, burdensome symptom among renal transplant recipients (RTR). According to the studies performed by our group, CKD-aI may persist after kidney transplantation (KTx) in 26.3% of cases, while 50% of itch in RTR appears after a successful KTx [8]. Moreover, CKD-aI is associated with decreased quality of life (QoL), a higher prevalence of depression and anxiety [9]. The pathophysiology of CKD-aI is not fully understood. Among possible pathogenetic mechanisms, the authors mention disturbances in peripheral opioid receptors expression, skin xerosis, microinflammation, hyperparathyroidism, and peripheral neuropathy [10,11,12,13].

It was already proven that interleukin-31 (IL-31), which is often called an itchy cytokine, plays an important role in the development of itch in multiple dermatological and systemic disorders [14,15,16,17,18,19,20]. Moreover, our group has recently described significant differences in IL-31 concentrations between itchy and non-itchy patients treated with HD [21]. 

Therefore, the aim of this study was to assess the differences in concentration of IL-31 among itchy RTR and to compare it with non-itchy RTR and healthy controls (HC). Moreover, a possible correlation of IL-31 level with itch severity was assessed. 

## 2. Materials and Methods

The study was performed in accordance with the Declaration of Helsinki and was approved by the Wroclaw Medical University Institutional Review Board (KB-750/2021). All the patients were included in the study after obtaining informed consent. 

### 2.1. Participants

The study was performed on a group of selected 129 RTRs (74 males, 57.4%, and 55 females, 42.6%) aged 52.01 ± 13.54 years old, who were under monthly supervision in the outpatient clinic of the Department of Nephrology and Transplantation Medicine of Wroclaw Medical University, Wroclaw, Poland, between March and November 2021. Adult patients with functioning renal transplants who were able to sign an informed consent were included in the study. Patients with a history of chronic dermatological disorders were excluded. Demographic data, including sex, age, BMI, disease duration, time on HD, time after KTx, atopic predisposition, and family history, were collected. Fifty-four (41.9%) patients (28 males and 26 females) aged 53.07 ± 12.0 years old reported suffering from itch in the previous three days. The patients suffered from CKD on average for 21.84 ± 13.01 years, spent 2.72 ± 2.11 years on HD, and were 8.7 ± 6.54 years after KTx. Among them, 9 (16.7%) reported atopic predisposition, while 11 (20.5%) had a positive family history of atopy. Out of all demographic data, only itch during HD was significantly more frequent in the itchy group (17.3% vs. 61.1%, respectively, *p* < 0.001). The patients’ characteristics are summarized in Table 1.

Moreover, 47 healthy volunteers from the Wroclaw Blood Donation Center were employed as a control group.

### 2.2. Itch

Itch intensity, in the previous three days before completing the questionnaire, was assessed using the worst itch numeral rating scale (WI-NRS) and the 4-Item Itch Questionnaire (4IIQ). The NRS is a commonly used instrument for determining itch severity on an 11-point scale, where 0 = no itch and 10 = worst imaginable itch [22]. The following cut of points were applied: 1–2 points represent mild itch, 3–6 points moderate itch, 7–8 points severe itch, and ≥9 points very severe itch [22]. 4IIQ is an instrument created and validated by our group which besides assessing itch severity on a 6-point scale (0–5 points), determines its frequency (0–5 points) and sleep impairment (0–6 points). The higher the score, the more severe the itch [23].

### 2.3. IL-31 Concentration

A total of 9 mL of blood was drawn from all participants during the routine monthly sampling. Then, the samples were centrifuged at 3000× *g* rpm for 15 min. Subsequently, the obtained serum was stored at—80 °C. The serum level of IL-31 was measured using the ELISA (enzyme-linked immunosorbent assay) technique using the commercially available Nori Human IL-31 ELISA Kit (catalog number: GR 111374, GENORISE SCIENTIFIC, Inc., Glen Mills, PA, USA), according to the manufacturer’s instructions. The absorbance of the sample was measured at 450 nm using the EPOCH (BioTEK^®^ Instruments, Inc., Winooski, VT, USA) adjustable microplate reader. IL-31 had a test range of 50–3200 pg/mL and a sensitivity of 10 pg/mL.

### 2.4. Statistical Analysis

Statistical analysis was performed using IBM SPSS Statistics v. 26 (SPSS Inc., Chicago, IL, USA). All data were assessed for normal distribution with the use of Shapiro–Wilk tests. The minimum, maximum, mean, and standard deviation numbers were calculated. Analyzed quantitative variables between two groups were compared using T-Student or Mann–Whitney U tests, while correlations were assessed using Spearman and Pearson coefficients. For the qualitative data test, Chi2 was used. Differences in IL-31 concentration between three groups were assessed with the use of the Kruskal–Wallis test or analysis of variance (ANOVA). A 2-sided *p* value ≤ 0.05 was considered to be statistically significant.

## 3. Results

Itch severity in the itchy RTR group was assessed as moderate with the mean NRS score of 4.98 ± 2.41 points, with no statistically significant difference between sexes. According to NRS cut-offs, most frequently patients reported itch of moderate-intensity (30 patients, 55.56%), followed by mild (10 patients, 18.52%), severe (9 patients, 16.67%), and very severe (5 patients, 9.25) itch. On average, patients scored 6.61 ± 2.51 points on 4IIQ, without significant differences between males and females. Regarding the IL-31 serum level, the mean concentration differed significantly between RTR suffering from itch (602.44 ± 534.5 pg/mL), non-itchy RTR (161.49 ± 106.61 pg/mL), and HC (110.33 ± 51.81 pg/mL) (*p* < 0.001) (Table 2). 

Post-hoc analysis revealed a statistically significantly increased IL-31 serum concentration in itchy RTR in comparison to the non-itchy RTR group (*p* < 0.001) and HC (*p* < 0.001). No significant difference was observed in IL-31 serum levels between non-itchy RTRs and HC (Figure 1). 

There was no correlation found between IL-31 concentration and severity of itch assessed both with NRS and 4IIQ (detailed data not shown). Moreover, no differences in IL-31 concentrations were found analyzing RTRs according to different demographic data and medical history.

## 4. Discussion

IL-31, often called an itchy or pruritogenic cytokine, is implicated in itch development in various skin disorders [24,25]. Its function was described for the first time in an induced mouse model of atopic dermatitis (AD) in 2004 [26] and since then its role in inflammation, tissue homeostasis, and pruritus has been vastly studied [27,28]. IL-31 is a T-cell-derived cytokine and belongs to the family of gp130/IL-6-derived cytokines. IL-31 expresses its effect through a heterodimeric receptor consisting of IL-31 receptor A (IL31RA) and Oncostatin M receptor (OSMR) [29]. Activation of the IL-31 receptor results in activation of various pathways, including JAK, STAT, or PI-3 kinase, and a subsequent affectation of a variety of cells, including epithelial cells, keratinocytes, peripheral sensory neurons, and the dorsal horn of the spinal cord [24]. In the skin, IL-31 inhibits keratinocyte differentiation by downregulation of fillagrin and involucrin expression, leading to an epidermal thickening and an increase in transepidermal water loss [30]. It was also observed that overexpression of IL-31 is associated with sensory neuronal outgrowth [31]. Scratching behaviors with induction of AD-like lesions were observed in IL-31 transgenic mice [26]. Moreover, continuous or intermittent injections of IL-31 induce scratching and development of dermatitis [32], whereas the administration of IL-31 antibodies diminishes such behaviors [33]. 

Multiple authors have reported increased serum protein and mRNA levels of IL-31 in adults and children with AD compared to HC [19,34,35]. Moreover, a decrease in IL-31 concentration was observed after treatment with an H1-histamine receptor antagonist and topical corticosteroids [36]. Nevertheless, the correlation of IL-31 levels with IgE and AD severity remains unconfirmed. While certain studies have identified its positive correlation with AD severity and IgE levels [17,19,20,37], other authors have denied such an association [20,35,38]. Similar observations were reported for other pruritic disorders. IL-31 levels are elevated in patients suffering from chronic prurigo, bullous pemphigoid, cutaneous T-cell lymphoma stasis dermatitis, dermatomyositis, and psoriasis [14,15,16,18,27,39]. 

So far, there are only a few published studies on the possible contribution of IL-31 in the pathogenesis of CKD-aI, and their results are not unequivocal. The first paper on the topic was published by Ko et al. [40] in 2014 on a group of 178 patients. The authors showed that patients suffering from CKD-aI had higher IL-31 serum levels than non-itchy HD patients. Moreover, the IL-31 level was considered an independent predictor for higher pruritus intensity [40]. Recently, our group has also shown a statistically significant increase in serum IL-31 levels in patients with CKD-aI on HD in comparison to non-itchy HD patients [21]. Similar results were reported by Haggag et al. [41] on a group of 88 patients with CKD. The authors have found that levels of IL-31 were not only higher among patients with itch on HD in comparison to non-itchy HD patients, but also in itchy patients with different degrees of CKD. Nevertheless, it is worth underlining that the results did not achieve statistical significance (*p* = 0.199). In the cross-sectional study by Oweis et al. [42], the authors compared 65 HD patients with 49 HC and found that HD patients had higher IL-31 levels. At the same time, the differences in the concentration of IL-13 and IL-33 were not statistically significant [42]. The sole analysis comparing IL-31 levels with RTR was published by Güvercin et al. [43] in 2019. This retrospective cohort study on 145 patients revealed no differences in IL-31 levels between patients on HD, peritoneal dialysis, RTR, and HC [43]. Nonetheless, the authors concluded that IL-31 might contribute to developing longitudinal nail ridges, as those patients had significantly higher serum levels of IL-31 [43]. The results of our study clearly show an important, statistically significant difference in IL-31 serum levels between itchy and non-itchy RTR. 

Nevertheless, due to the differences in study design, the lack of inclusion of RTR in previous studies, or the lack of distinction between itchy and non-itchy RTR, it is challenging to extrapolate and compare our results with the already published data. Still, it is essential to emphasize that a clear trend of high IL-31 concentrations in itchy patients with any CKD stage may be caused by the involvement of IL-31 and the dysregulation of the immune system in developing the CKD-aI. This hypothesis has already been proposed by Kimmel et al. [44], who believed that the continuous state of microinflammation shifts the Th1 differentiation and production of pro-inflammatory cytokines in CKD-aI patients. It was subsequently confirmed by showing higher IL-2, IL-6, and C-reactive protein concentrations in itchy CKD patients [44,45,46].

Itch severity and its correlation with IL-31 levels have already been studied in CKD patients, yet the available results remain ambiguous. Ko et al. [40] found a statistically significant, yet weak (r = 0.15), correlation between IL-31 levels and the visual analog scale (VAS) of pruritus intensity. This is the only study indicating such a correlation in CKD-aI patients. In the study by our group [21], a positive trend of moderate strength was observed. However, it did not reach statistical significance (*p* = 0.058). Similarly, Oweis et al. [42] reported a statistically significant relationship between IL-13 and the itch score assessed with the pruritus grading system, yet no correlation for IL-31 was observed. Correspondingly, among RTR patients in our study, no statistical correlation with itch severity assessed with both NRS and 4IIQ was observed.

It is essential to underline the practical aspect of our study. It has already been proven that CKD-aI is associated with a critical burden and a marked decrease in various health-related QoL domains [47]. CKD-aI is associated with lower QoL, decreased sleep quality, lower functional capacity, decreased relationship satisfaction, higher prevalence of depressive symptoms, anxiety, and alexithymia [48,49,50,51,52]. Moreover, the worsening of many symptoms mentioned above is associated with itch intensity [48,49,50,51,52]. Our group made similar observations on itchy RTR earlier this year [9]. CI was associated with a 3.7-fold increase in the prevalence of depression and a 4.8-fold increase in the prevalence of anxiety. Moreover, positive, moderate to strong, significant correlations were found between ItchyQoL and both anxiety and depression [9]. Moreover, it was proven that the presence of CKD-aI among HD patients may be associated with a 17% higher mortality rate [6].

The efficacy of anti-IL-31RA (nemolizumab) treatment has been demonstrated for the treatment of itch in atopic dermatitis and chronic prurigo [53,54]. According to the meta-analysis performed by Liang et al. [55] nemolizumab is a promising drug for inadequately controlled AD patients. Nemolizumab significantly reduces the itch, as well as AD severity compared to placebo [55]. Moreover, the results from two phase III long-term studies in patients with atopic dermatitis have confirmed that the use of nemolizumab with topical treatments produces even greater, continuous improvement in pruritus, signs of the disease, and patients’ QoL [56]. Similar findings were observed in the treatment of chronic prurigo. The use of nemolizumab resulted in a greater reduction in itch and severity of skin lesions [54]. At present, there is only one phase II study on the efficacy of anti-IL-31RA in CKD-aI patients published [57]. Unfortunately, the efficacy of the drug was not observed, although patients with higher serum IL-31 at screening tended to have a greater decrease in itch intensity [57]. Nevertheless, future double-blind clinical trials with different blinding conditions and improved demographical balance could fully unveil the efficacy of nemolizumab in CKD-aI. 

We acknowledge that our study has some limitations. Firstly, we analyzed only serum IL-31 levels without measuring the expression of mRNA of IL-31 and its receptor in the skin. Moreover, we did not correlate IL-31 with other inflammatory interleukins and possible factors of CKD-aI (including parathormone and CRP). Future studies in this area will be necessary to fully understand the role of IL-31 in the pathogenesis of CI in RTR.

## 5. Conclusions

In conclusion, to the best of our knowledge, this is the first study assessing the disturbances in serum IL-31 concentrations in itchy RTR. The results of our study clearly demonstrate the association between IL-31 levels and CI in patients after renal transplantation. This relationship was previously observed for HD patients and therefore one may think of a possible mutual pathomechanism of CI. Hopefully, future studies targeting this cytokine will bring important information necessary for the introduction of novel therapeutic options for CI in RTRs.

## Figures and Tables

**Figure 1 jcm-11-04309-f001:**
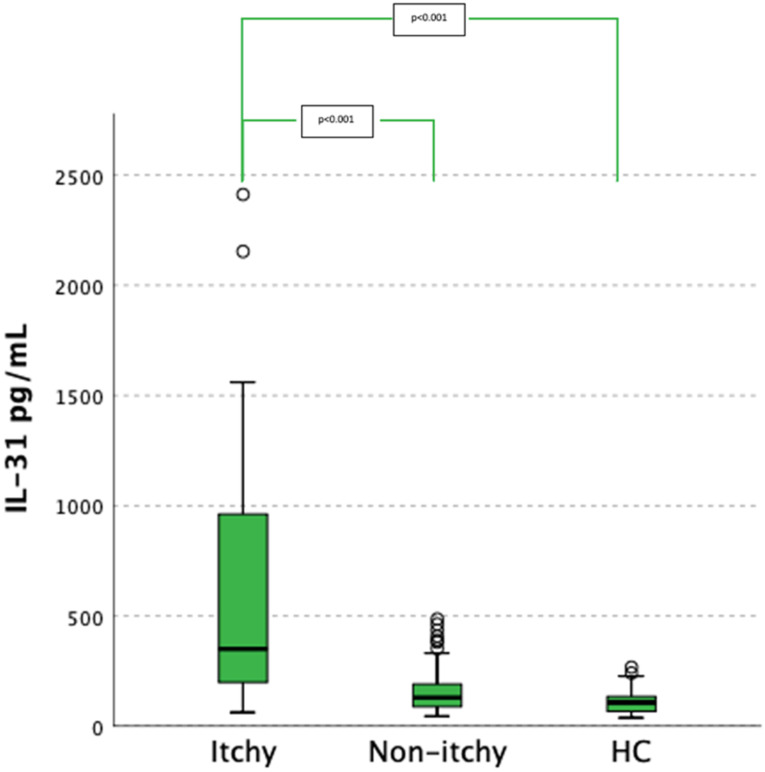
Kruskal–Wallis test and posthoc analysis of the differences in serum IL-31 concentrations between studied groups. HC—healthy controls.

**Table 1 jcm-11-04309-t001:** Patients’ demographics.

Characteristics	No Itch in Last 3 Days (*n* = 75)	Itch in Last 3 Days (*n* = 54)	*p*
Sex, *n* (%)			NS
Male	46 (61.3)	28 (51.9)	
Female	29 (38.7)	26 (48.1)	
Age (years, mean ± SD)	51.24 ± 14.58	53.07 ± 12.0	NS
BMI (kg/m^2^, mean ± SD)	25.7 ± 5.5	26.22 ± 5.06	NS
Time of disease (years, mean ± SD)	18.47 ± 11.51	21.84 ± 13.01	NS
Time on dialysis (years, mean ± SD)	2.13 ± 1.84	2.72 ± 2.11	NS
Time after KTx (years, mean ± SD)	7.1 ± 7.45	8.7 ± 6.54	NS
Atopy, *n* (%)	12 (16)	9 (16.7)	NS
Atopy in family *n* (%)	11 (14.7)	11 (20.4)	NS
Itch on dialysis *n* (%)	13 (17.3)	33 (61.1)	<0.001
Itch (points, mean ± SD)	NA	4.98 ± 2.41	NA

*n*—number of patients; SD—standard deviation; BMI—body mass index; KTx—kidney transplantation, NA—not applicable, NS—not significant.

**Table 2 jcm-11-04309-t002:** Differences in IL-31 concentrations between studied groups.

Characteristics	No Itch in Last 3 Days (*n* = 75)	Itch in Last 3 Days (*n* = 54)	Healthy Controls	*p*	Post-Hoc
IL-31 level (pg, mean ± SD)	161.49 ± 106.61	602.44 ± 534.5	110.33 ± 51.81	<0.001	Itch vs. no itch < 0.001 Itch vs. control < 0.001 No itch vs. control 0.058

SD—standard deviation; *n*—number of patients.

## Data Availability

Available on request from the corresponding author.

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
