# Peer review of "High Serum IL-31 Concentration Is Associated with Itch among Renal Transplant Recipients"

_jcm, 2022, doi:10.3390/jcm11154309_

Round 1
Reviewer 1 Report
This is a cleverly designed and well presented study on a topic of wide interest. The results are strongly suggestive of need for further research on the intricate relationship of IL 31 and itch. The paragraph on limitations of the current study adds to manuscript integrity.
Author Response
Dear Reviewer,
Thank you for the time and possibility to review our manuscript. We are thankful that you think so highly about our work.
Reviewer 2 Report
In this study, Piotr et al tested the concentration of IL-31 among itchy and non-itchy renal transplant recipients (RTR). They showed that the concentration of IL-31 is significantly different between itchy and non-itchy RTR. This finding opens a new path for us to treat itch in RTR. The authors should clarify why IL-31 rather than other cytokines was chosen for testing. The role of IL-31 in itch diseases has been reported by some groups. The author should state the function of IL-31 in itch sensation in the introduction.
Author Response
Dear Reviewer,
Thank you for the time and possibility to review our manuscript. We are thankful that you think so highly about our work.
We have added a short paragraph in the Introduction regarding IL-31. We agree that it will improve the manuscript and explain its background. It was highlighted in yellow
Reviewer 3 Report
This manuscript "High serum IL-31 concentration is associated with itch among renal transplant recipient," is both interesting and informative. I enjoyed reading it. This study of 129 renal transplant recipients from a nephrology and transplantation medicine university clinic provides valuable information. Interleukin-31 is implicated in itch development
in a number of cutaneous disorders. The efficacy of subcutaneous administered nemolizumab, a humanized monoclonal anti-IL-31-receptor alpha antibody, as a therapeutic option for pruritus in atopics further highlights the significance of studying serum IL-31 concentrations.
Pruritus is an important topic and a distressing symptom.
The value of anti-IL-31-receptor alpha antibody as a option for the common disorder of atopic dermatitis is relevant and important.
Accordingly, documenting elevated serum IL-31 concentrations in previously unstudied category, transplant recipients with pruritus, is salient. In addition, studying a large number of patients and being able to utilize a nephrology and transplantation medicine university clinic for this study makes it truly remarkable.
This study is unique in that it evaluated renal transplant recipients suffering from pruritus rather than a population of atopics. In addition, studying a large number of patients and being able to utilize a nephrology and transplantation medicine university clinic for this study makes it truly remarkable.
This study is unique in that it evaluated 129 renal transplant recipients from a nephrology and transplantation medicine university clinic, an important population often suffering from pruritus.
It is well written, especially for authors whose first language is unlikely to be English.
This important work adds further on an important yet unstudied population, renal transplant recipients, concerning the status of serum IL-31 concentration in those suffering from pruritus. In addition, studying a large number of patients and being able to utilize a nephrology and transplantation medicine university clinic for this study makes it truly remarkable.
Author Response

(The authors gave the same response as above.)

Reviewer 4 Report
This is an excellent paper about the role of IL-31 serum levels and it´s correlation with itch in renal transplant patients. The introduction is sound, the methods clearly described, discussion includes the limitation of the study and the literature is well cited in the field of IL-31.
I have just one minor comment if it would be possible to include the itch score of NRS in the patients characteristics.
Author Response
Dear Reviewer,
Thank you for the time and possibility to review our manuscript. We are thankful that you think so highly about our work.
We have included the mean +/- SD NRS score for pruritus in the Table. It was highlighted in yellow